# Tuning Photothermal Properties of Graphene Oxide by Heteroatom Doping for Cancer Elimination: Experimental and DFT Study

**DOI:** 10.3390/ijms262411771

**Published:** 2025-12-05

**Authors:** Alan Miranda, Mateo Salazar, D. G. Larrude, Leandro Seixas, Alexis Debut, Myriam González, Karina J. Lagos, Orlando Campaña, Miryan Rosita Rivera, Maria Paulina Romero

**Affiliations:** 1Escuela Politécnica Nacional, Quito 170525, Ecuador; alankonker98mo@gmail.com (A.M.); myriam.gonzalez01@epn.edu.ec (M.G.); kari_jann@hotmail.com (K.J.L.); kleber.campana@epn.edu.ec (O.C.); 2Laboratorio de Investigación en Citogenética y Biomoléculas de Anfibios (LICBA), Centro de Investigación para la Salud en América Latina (CISeAL), Facultad de Ciencias Exactas y Naturales, Pontificia Universidad Católica del Ecuador (PUCE), Quito 170143, Ecuador; massuno@hotmail.com (M.S.); mriverai@puce.edu.ec (M.R.R.); 3Graphene and Nano-Materials Research Center—MackGraphe, Mackenzie Presbyterian University, São Paulo 01302-907, Brazil; dunieskys@gmail.com; 4Instituto de Fisica Teórica, Universidade Estadual Paulista—Júlio de Mesquita Filho, São Paulo 01140-907, Brazil; seixasle@gmail.com; 5Centro de Nanociencia y Nanotecnología, Universidad de Las Fuerzas Armadas ESPE, Sangolquí 171103, Ecuador; apdebut@espe.edu.ec

**Keywords:** cancer, photothermal therapy, graphene oxide, doping, photosensitizing agents, computational simulations

## Abstract

Cancer poses a global challenge, affecting millions of people and placing a significant burden on families and healthcare systems. Chemotherapy, radiotherapy, hormone therapy, and immunotherapy are commonly used for cancer treatment; their side effects can be severe. Photothermal therapy (PTT) has emerged as a promising alternative due to its minimal invasiveness and high efficiency. In this study, graphene oxide (GO) was synthesized and functionalized to obtain nitrogen-doped graphene oxide (NGO) and boron-doped graphene oxide (BGO) via a hydrothermal process, aiming to use them as photoactive agents (PAs) in PTT. Atomic force microscopy (AFM) analysis revealed that GO, BGO, and NGO exhibit monolayer atomic structures. Spectroscopic analyses confirmed the presence of oxygen and carbon in all samples, along with successful boron and nitrogen doping in BGO and NGO, respectively. Cytotoxicity assays yielded half-maximal inhibitory concentrations (IC_50_) of 1025.26 μg/mL for GO, 2695.03 μg/mL for BGO, and 1319.81 μg/mL for NGO. Photothermal experiments were conducted using a 635 nm light source with an intensity of 65.5 mW/cm^2^, resulting in temperature thresholds of 44.87 °C for GO, 48.36 °C for NGO, and 55.91 °C for BGO. Anticancer assays were performed using the T-47D breast cancer cell line, demonstrating tumor cell elimination rates of 97.93% for GO, 98.54% for BGO, and 97.98% for NGO, underscoring their efficacy as PAs. Density functional theory (DFT) simulations were carried out to determine the absorbance coefficient as a function of doping percentage. The results revealed that increased doping enhances light absorbance and, consequently, the photothermal response, as higher absorbance at the irradiation wavelength leads to greater energy absorption and temperature elevation.

## 1. Introduction

Cancer carries a high mortality rate globally, evidenced by the registration of 1,958,310 new cases in 2023 and an estimated 609,820 deaths attributable to this disease in the United States alone [1]. Therefore, seeking more efficient ways to combat this affliction is imperative. There exists a wide array of conventional cancer treatments, such as chemotherapy, radiotherapy, or immunotherapy [2], yet these methods entail detrimental side effects for the patient. Chemotherapy, for instance, increases susceptibility to infections and conditions like anemia due to cytotoxic agents affecting both tumor and healthy cells [3], thereby impairing the endogenous antitumor response of immune cells that may be present [4]. Radiotherapy shows skin reactions and the development of secondary cancers [5], while immunotherapy induces organ inflammation and hormonal imbalances [6].

To improve the quality of life of cancer patients, new therapeutic alternatives have been proposed to complement or replace conventional treatments. PTT offers significant advantages, such as fewer adverse effects. Unlike traditional methods such as chemotherapy or radiotherapy, PTT typically causes only mild discomfort in the treated area [7], superficial skin reactions, or minimal inflammatory responses [8], making it a safer and more tolerable option for patients.

PTT is a minimally invasive treatment that selectively targets and destroys tumor cells while minimizing damage to surrounding healthy tissues [9]. PTT involves transporting PAs (nanomaterials capable of converting light energy into heat) to the tumor site, subsequently irradiating these regions with electromagnetic waves, thereby increasing temperature and inducing cancer cell death through hyperthermia [9].

The wavelengths that penetrate skin tissue depend on skin type, thickness, and skin-specific chromophores like hemoglobin and melanin [10]. The penetration depth of the light into the skin increases depending on the wavelength of the light, ranging from ultraviolet to visible light and infrared [11]. Thus, red and infrared light are utilized in PTT for tumor elimination because they penetrate deeper into the skin layers than blue light [9].

Carbon-based materials are of great interest for use as PAs due to their high biocompatibility, excellent thermal conductivity, and large specific surface area [12]. Moreover, they do not exhibit cytotoxic responses [13], unlike other nanomaterials such as silver nanoparticles, which induce mitochondrial dysfunction, reactive oxygen species that damage DNA, and chromosomal aberrations [14]; cadmium selenide quantum dots that affect cell osmotic balance [13]; and silica nanoparticles that induce oxidative stress and mitochondrial apoptosis [15].

One of the primary carbon-based materials used in PTT is GO [16,17], mainly due to its strong absorbance in the infrared region [17] and its straightforward and diverse synthesis methods like the Staudenmaier [18] or Hummers [19] methods. There is ongoing interest in chemically modifying GO by doping it with different elements to enhance its response to electromagnetic wave irradiation [20,21,22], thereby improving PTT efficiency. Among the prominent dopants for GO are nitrogen doping [18,23] and boron doping [24], since nitrogen and boron have atomic numbers higher and lower than carbon, respectively, facilitating substitution of a carbon atom in the GO structure [24,25]. It has been demonstrated that modifying GO with nitrogen and boron enhances its ability to convert light energy into thermal energy [21,22,23], making them promising candidates as PAs in PTT.

This study aimed to synthesize GO using the modified Hummers method and to functionalize it with nitrogen and boron to obtain NGO) and BGO. The physical and chemical properties of the samples were evaluated using Fourier Transform Infrared Spectroscopy (FTIR), Raman Spectroscopy, X-ray Photoelectron Spectroscopy (XPS), Scanning Electron Microscopy (SEM), and AFM. Additionally, cytotoxicity tests were conducted via live cell counting to determine the concentration at which the material can have biological applications, along with anticancer assays to demonstrate the efficiency of the nanomaterials used as PAs in PTT. A photothermal test was performed to evaluate the photothermal properties of the material by irradiating it with a light source at a wavelength of 635 nm and a light dose of 65.5 mW/cm^2^, determining the maximum temperature the samples can reach due to irradiation. Finally, computational simulations based on DFT were carried out to elucidate how the photothermal properties vary with structural modifications of the samples, demonstrating changes in material absorbance based on the chosen doping type and subsequent variations in the maximum temperature achievable upon light irradiation.

## 2. Results

### 2.1. Characterization

#### 2.1.1. Characterization of GO, NGO, and BGO Morphology

GO, BGO, and NGO morphology was characterized using SEM. Figure 1a–c shows a micrograph of GO, BGO, and NGO, respectively, revealing the characteristic sheet-like morphology of graphene [26].

The thickness of the samples was determined using AFM in tapping mode. For GO, BGO, and NGO, the height was measured in five different regions, marked in Figure 1 with colored lines. The results are presented in Figure 2a–f for GO, BGO, and NGO, respectively. The average thicknesses for the GO, BGO, and NGO samples are 1.38 nm, 1.58 nm, and 1.37 nm, respectively. Additionally, based on the average heights recorded, it was determined that GO, BGO, and NGO are monolayers [27].

#### 2.1.2. Characterization of GO, NGO, and BGO Chemistry Structure

FTIR was performed to identify the functional groups in GO, BGO, and NGO. Figure 3 presents the corresponding FTIR spectra for each sample.

Figure 3a shows the FTIR spectrum of pure GO, where several characteristic bands associated with oxidation are observed. The broad band around 3340 cm^−1^ corresponds to the stretching vibration of hydroxyl groups (O–H) [28]. A prominent band at 1730 cm^−1^ is attributed to carbonyl groups (C=O), while the band at 1625 cm^−1^ is related to stretching unoxidized carbon-carbon double bonds (C=C). Additionally, bands at 1410 cm^−1^ and 1223 cm^−1^ are assigned to epoxy groups (C–O) [29].

Figure 3b displays the FTIR spectrum of BGO. Similarly to GO, it exhibits bands related to O–H, C=O, C=C, and C–O functional groups. These bands appear at 3260 cm^−1^, 1703 cm^−1^, 1560 cm^−1^, and 1410 cm^−1^, respectively [30]. Moreover, a distinct band at 1040 cm^−1^, characteristic of boron–carbon (B–C) bonds [26], confirms successful boron doping.

Figure 3c shows the FTIR spectrum of NGO. Oxidation-related bands are also present, with the O–H, C=O, C=C, and C–O stretching vibrations observed at 3004 cm^−1^, 1730 cm^−1^, 1567 cm^−1^, and 1032 cm^−1^, respectively [26]. In addition, a distinctive band at 1187 cm^−1^ corresponds to nitrogen–carbon (N–C) bonds [26], indicating the incorporation of nitrogen atoms into the graphene structure.

Overall, all samples exhibit functional groups commonly associated with oxidized graphene. However, the presence of the B–C and N–C characteristic bands in BGO and NGO provides clear evidence of successful doping with boron and nitrogen atoms.

Raman spectroscopy is a powerful, non-destructive technique widely used to characterize carbon-based materials, particularly graphene and its derivatives. It provides detailed information about the vibrational modes of a material, allowing the identification of structural features such as the degree of disorder, the presence of functional groups, and the number of graphene layers. The most significant Raman bands, such as the D, G, and 2D bands, serve as fingerprints for evaluating the quality and structural integrity of graphene-based samples [27]. Figure 3d–f shows the Raman spectrum for GO, BGO, and NGO.

The Raman spectra of GO, BGO, and NGO were deconvoluted using Lorentzian fitting functions. This allowed us to accurately extract the central positions, full width at half maximum (FWHM), and intensities of the D, G, 2D, D′, and additional bands observed. Lorentzian profiles were chosen because they best represent the phonon scattering processes in graphene-based materials, especially in cases involving structural disorder or doping [31]. The reliability of the deconvolution process was confirmed by the goodness-of-fit (R^2^ > 0.99) for all analyzed peaks.

The ID/IG intensity ratio was calculated for all samples, yielding values of 1.55 for GO, 1.73 for BGO, and 1.54 for NGO. This ratio is below 3.5 in all three cases, indicating a high degree of crystallinity in the samples [27].

The Raman spectra of GO, BGO, and NGO are presented in Figure 3d–f. In all cases, the G band, associated with sp^2^-hybridized carbon atoms, is observed at 1570.54 cm^−1^ for GO, 1593.89 cm^−1^ for BGO, and 1550.88 cm^−1^ for NGO, respectively [27]. Likewise, the 2D band appears at 2742.08 cm^−1^, 2738.92 cm^−1^, and 2795.21 cm^−1^ for GO, BGO, and NGO, respectively [27]. These bands are characteristic of carbon-based materials and follow the Raman selection rules [27].

An additional analysis of the G and 2D bands reveals evidence of doping effects in the BGO and NGO samples. The blue shift in the G band in BGO (from 1570.54 to 1593.89 cm^−1^) is consistent with p-type doping, while the red shift in the G band in NGO (to 1550.88 cm^−1^), accompanied by a blue shift in the 2D band (to 2795.21 cm^−1^), suggests n-type doping. These trends align with previous reports on the Raman response of doped graphene, in which charge-transfer effects from electron-donating or withdrawing dopants alter phonon frequencies [22,32]. This behavior supports the successful incorporation of boron and nitrogen atoms into the graphene lattice, with distinct electronic effects depending on the dopant nature.

In addition, the presence of structural defects leads to the emergence of the D and D′ bands, located at 1350.5 cm^−1^ and 1608.25 cm^−1^ for GO, 1340.92 cm^−1^ and 1593.89 cm^−1^ for BGO, and 1347.41 cm^−1^ and 1700.13 cm^−1^ for NGO, respectively [27]. Another band identified around 3173.39 cm^−1^, 3166.11 cm^−1^, and 3178.73 cm^−1^ for GO, BGO, and NGO, respectively, corresponds to the 2D′ band, which is associated with second-order phonon scattering processes [28].

Additionally, new bands denoted as D* and D″ are observed between the D and G bands at 1141.32 cm^−1^ and 1533.48 cm^−1^ for GO, 1106.60 cm^−1^ and 1593.89 cm^−1^ for BGO, and 1347 cm^−1^ and 1550.89 cm^−1^ for NGO, respectively [27]. These bands have been linked to additional vibrational modes induced by structural defects or chemical functionalization [27].

Another noteworthy feature is the appearance of the G* band, located at 2663.52 cm^−1^, 2646.59 cm^−1^, and 2669.59 cm^−1^ for GO, BGO, and NGO, respectively. This band has not been previously reported in unpurified samples, suggesting that chemical treatment and doping influence the graphene structure [27].

Finally, all samples exhibit the D+D′ combination band, found at 2934.26 cm^−1^, 2925.99 cm^−1^, and 2933.82 cm^−1^ for GO, BGO, and NGO, respectively. This band can be activated even in structures with low defects, reflecting complex interactions between the material’s vibrational modes [27].

The XPS spectra of GO, NGO, and BGO are presented in Figure 4, Figure 5 and Figure 6. In all cases, two main bands are observed in the survey spectra: the C 1s band around 284 eV and the O 1s band near 532 eV, corresponding to the predominant elements in graphene oxide derivatives.

For GO (Figure 4), the C 1s spectrum can be deconvoluted into four components associated with different functional groups: C–C at 284.5 eV, C–O at 286.4 eV, C=O at 286.95 eV, and O–C=O at 287.85 eV [29]. The O 1s bands split into three contributions attributed to C=O, C–OH, and C–O–C groups, located at 531.507 eV, 532.407 eV, and 533.107 eV, respectively [29]. Elemental analysis shows an atomic composition of 65.53% carbon and 27.65% oxygen in the GO sample.

In the case of NGO (Figure 5), the C 1s bands are deconvoluted into three components: C–C at 284.48 eV, C–O at 286.48 eV, and O–C=O at 288.039 eV [22]. The O 1s region also reveals three bands corresponding to C=O (531.089 eV), C–OH (532.389 eV), and C–O–C (533.389 eV) [22]. In the survey spectrum (Figure 3a), a small band is observed around 400 eV, attributed to N 1s, confirming nitrogen incorporation. This N 1s region is deconvoluted into three bands corresponding to pyridinic (398.789 eV), pyrrolic (399.689 eV), and quaternary nitrogen (400.789 eV) [22]. The atomic composition of NGO is 74.15% carbon, 20.44% oxygen, and 3.92% nitrogen.

For BGO (Figure 6), the C 1s band shows four components located at 285.0 eV (C–C), 285.706 eV (C–O), 287.156 eV (C=O), and 288.256 eV (O–C=O) [30]. The O 1s bands are deconvoluted into two signals related to C=O at 531.656 eV and C–OH at 532.906 eV [30]. Similarly to NGO, the N 1s bands appear in the survey spectrum and are deconvoluted into pyridinic (399.278 eV), pyrrolic (400.517 eV), and quaternary (400.982 eV) contributions [22]. Additionally, a band at 192 eV is assigned to B 1s, confirming the presence of boron in the BGO sample. The B 1s spectrum splits into bands associated with BC_3_ and BC_2_O bonding configurations, located at 193.056 eV and 192.356 eV, respectively [30]. The atomic composition of BGO is 68.98% carbon, 24.45% oxygen, 2.05% nitrogen, and 4.52% boron.

### 2.2. GO, NGO, and BGO-Mediated Photothermal Therapy

The cytotoxic effects of GO, NGO, and BGO were evaluated on T-47D breast cancer cells after 24 h of incubation. Figure 7 presents the results of the cytotoxicity assays along with the corresponding dose–response curve fits. Based on these curves, IC_50_, defined as the concentration of a compound required to reduce cell viability by 50%, was determined for each material. The IC_50_ values were 1025.26 µg/mL for GO, 1319.81 µg/mL for NGO, and 2695.03 µg/mL for BGO, respectively. These values also provide insight into the potential anti-cancer properties of the studied materials.

Figure 8a presents the results of the anticancer assays based on PTT using the T-47D breast cancer cell line. The concentrations used 1000 µg/mL for GO and NGO and 2000 µg/mL for BGO were selected from the region below the IC_50_ values obtained in the cytotoxicity assays. This ensured that the compounds alone would not cause excessive toxicity, allowing the photothermal effect to be evaluated independently of the toxicity. Samples were incubated with the nanomaterials for 3 h before irradiation, while cytotoxicity analysis was performed after 24 h of exposure.

As shown in Figure 8a, no significant difference was observed between the control groups under light or dark conditions, both of which maintained nearly 100% cell viability, confirming that light alone does not induce cytotoxicity. In contrast, the dark (non-irradiated) samples treated with GO, BGO, and NGO exhibited moderate reductions in cell viability—55.63%, 61.99%, and 49.27%, respectively—reflecting some baseline cytotoxicity. However, upon irradiation, a dramatic decrease in viability was observed: 2.07% for GO, 1.46% for BGO, and 2.017% for NGO, confirming a strong photothermal effect. These differences were statistically significant compared to the control and dark-treated groups (*p* ≤ 0.001, Tukey’s test).

Photothermal studies were conducted to evaluate the heating and relaxation behavior of GO, NGO, and BGO under near-infrared irradiation, using water as a reference. Figure 8b displays the resulting temperature profiles along with their fitted curves. The threshold temperatures achieved by GO, NGO, BGO, and water were 44.87 °C, 48.36 °C, 55.91 °C, and 33.67 °C, respectively. The corresponding relaxation times were 4.35 min for GO, 2.21 min for NGO, 2.53 min for BGO, and 7.45 min for water.

The relaxation time is the time it takes for the material to cool after stopping light irradiation. A shorter relaxation time indicates that the material dissipates heat rapidly, while a longer time reflects greater thermal retention. In this study, NGO and BGO exhibited faster relaxation times than GO and water, suggesting a more dynamic heat-release process that can help limit unwanted thermal damage to surrounding tissues after treatment.

Overall, BGO demonstrated the most effective photothermal behavior, achieving the highest temperature and exhibiting rapid yet controlled thermal relaxation. These features highlight BGO as a highly promising photothermal cancer therapy agent capable of both strong heating and efficient post-irradiation cooling.

Atomistic simulations.

Before presenting the computational results, it is important to clarify a methodological distinction: the experimental investigations in this study were conducted on NGO and BGO, whereas the atomistic simulations were performed on pristine graphene lattices doped with nitrogen or boron, without explicitly including oxygen functionalities. Consequently, the simulations are not intended to provide a direct quantitative reproduction of the experimental data, but rather to offer qualitative insights into how substitutional doping modifies the electronic structure—such as Fermi-level shifts, variations in the density of states, and changes in optical absorption—that underlie the observed photothermal behavior.

To support the experimental findings and provide insight into the electronic properties that influence photothermal behavior, atomistic simulations were performed using DFT, as implemented in the SIESTA code [33,34]. An orthorhombic supercell containing 60 carbon atoms was used to model pristine graphene. One carbon atom was substituted for a nitrogen or boron atom in doped structures.

The simulation framework established here serves as a basis for analyzing how atomic-scale modifications, such as doping, can influence electronic transitions and enhance photothermal behavior, providing a theoretical foundation for interpreting experimental photothermal results.

The simulations employed Troullier-Martins norm-conserving pseudopotentials, an energy cutoff of 500 Ry, and a double-ζ polarized (DZP) basis set with a confinement energy cutoff of 0.02 Ry. The PBEsol exchange-correlation functional was selected, and all structures were optimized until forces were below 0.01 eV/Å. A vacuum spacing of 20 Å was included along the z-direction to avoid interlayer interactions. Brillouin zone sampling was performed using a 4 × 4 × 1 Monkhorst-Pack k-point mesh, and optical properties were calculated using a 50 × 50 × 1 q-point mesh with Gaussian broadening of 0.1 eV.

Figure 9 shows the electronic band structure of pristine graphene, which displays a linear dispersion at the K point of the Brillouin zone, characteristic of massless Dirac fermions. These fermions are responsible for graphene’s exceptional electronic properties, including high charge carrier mobility, broadband light absorption, and rapid electron relaxation dynamics. These features make graphene and its derivatives particularly suitable for photothermal conversion, as they facilitate efficient absorption of incident light and its conversion to heat [32].

Figure 10a–c shows the band structure and PDOS for NGO, corresponding to doping concentrations of 1.67%, 5%, and 6.67%. As the nitrogen concentration increases, a noticeable upward shift in the Fermi level is observed. This shift indicates nitrogen atoms donate electrons to the graphene structure, enhancing its n-type character. The Dirac point is displaced below the Fermi level, and the PDOS becomes more asymmetric near the Fermi energy, reflecting an increase in available electronic states for conduction.

Figure 10d–f displays the corresponding band structure and PDOS for BGO at the same doping levels: 1.67%, 5%, and 10%, labeled. In this case, the Fermi level shifts downward as the boron concentration increases due to boron acting as an electron acceptor. This introduces a p-type behavior in the graphene, with the Dirac point moving above the Fermi level. The density of states near the Fermi level also becomes more pronounced, indicating greater localization of states that can participate in optical and thermal interactions.

Overall, nitrogen doping gradually increases the Fermi level, while boron doping leads to a decrease. These opposite trends are consistent with nitrogen and boron’s donor (n-type) and acceptor (p-type) roles, respectively. These electronic structure modifications are directly linked to the photothermal properties of the materials, as doping alters the distribution of energy levels and electronic transitions, thereby enhancing light absorption and conversion efficiency.

Figure 11a,c presents the absorbance coefficient of graphene doped with nitrogen and boron over a wavelength range of 200 nm to 900 nm. In both cases and for all doping percentages analyzed, a higher intensity band is observed around 290 nm, associated with π–π* transitions of the carbon network [35].

On the other hand, Figure 11b,d display the absorbance coefficient in the range of 500 nm to 2400 nm. The absorbance coefficient increases with increasing doping percentage. For doping at 1.67%, the absorbance coefficient decreases compared to pristine graphene. However, for 3.33% doping, this coefficient increases with the wavelength, reaching levels similar to pristine graphene and then surpassing it at higher doping, especially around 1425 nm, as seen in Figure 11b.

The atomistic simulations conducted in this work (Figure 12) offer key insights into the origin of the enhanced photothermal behavior observed experimentally for doped graphene oxides, particularly BGO. The results, illustrated in the absorbance coefficient plots (Figure 11a–d), are directly connected to the experimental T vs. t profiles, where BGO exhibited the highest threshold temperature and efficient heat relaxation, confirming its superior photothermal conversion capacity.

The band-structure calculations revealed that both nitrogen and boron doping significantly shift the Fermi level. For the NGO, an upward shift in the Fermi level was observed with increasing nitrogen content, reflecting n-type behavior due to the donor nature of nitrogen. In contrast, BGO showed a downward shift in the Fermi level, indicating a p-type character from the electron-acceptor nature of boron. These shifts modify the carrier concentration and can enhance light-matter interactions, which are critical in photothermal conversion.

The PDOS analysis confirmed that doping introduces new electronic states near the Fermi level, enhancing the material’s ability to absorb light. The increase in available states facilitates more efficient excitation of electrons upon photon absorption, a key step in converting light energy into heat. BGO showed broader, more intense features near the Fermi level at higher doping concentrations, consistent with its better photothermal response.

Furthermore, the optical absorption spectra of doped graphene structures showed increased absorbance in the visible and NIR regions with increasing doping percentage.

In summary, the DFT simulations demonstrated that boron doping effectively tailors the electronic and optical properties of GO by (1) shifting the Fermi level, (2) introducing additional states near it, and (3) enhancing light absorption in the NIR region. These features correlate well with the experimentally observed higher photothermal efficiency of BGO, confirming that its electronic structure is optimized for light-to-heat conversion, making it a promising photothermal agent for cancer therapy.

It is important to note that while the experimental results were obtained using NGO and BGO, the atomistic simulations were performed on doped pristine graphene structures, without explicitly including oxygen-containing functional groups. As such, the comparison between theoretical and experimental data is intended to be qualitative, focusing on trends such as Fermi level shifts and absorbance variations as a function of doping. Including oxygen functionalities in future simulations could provide a more accurate representation of the experimental materials and further refine the correspondence between theory and experiment.

## 3. Discussion

FTIR and XPS Spectra confirmed the presence of carbon and oxygen in all three studied samples. The FTIR spectrum shown in Figure 3 reveals the bands corresponding to the C=C band, characteristic of the carbon framework structures [36]. XPS spectra of the C1s band for GO, BGO, and NGO also show a band associated with the C–C group, corresponding to the carbon skeleton [37]. Furthermore, the XPS spectra of the C1s band in GO, BGO, and NGO, as well as the O1s band, reveal the presence of C–O, C=O, O–C=O, C–OH, and C–O–C groups related to the oxidation of the carbon structure [27]. The FTIR spectra also show bands related to C=O and C–O groups, further confirming the oxidation of the studied materials [36]. These findings confirm the presence of functional groups indicative of carbon oxidation, validating that the synthesis process achieved material oxidation.

The atomic percentage analysis of the XPS spectra shows similar oxygen percentages in GO, BGO, and NGO, confirming that the functionalization process did not result in a significant loss of oxygen functional groups. Additionally, AFM micrographs reveal monolayers in GO, BGO, and NGO. The height profiles show uniform layer thicknesses across the samples, characteristic of graphene oxide.

Spectroscopic analysis of the NGO sample confirms the presence of nitrogen. The FTIR spectrum shows a band corresponding to the C–N group [38], while the XPS spectrum of the N1s band reveals the pyridinic, pyrrolic, and quaternary nitrogen bonding configurations in the graphene network [39]. Additionally, the atomic percentage analysis indicates 3.92% nitrogen in NGO. Therefore, it can be concluded that the employed methodology effectively incorporated nitrogen atoms into the graphene oxide network, resulting in successful doping. Spectroscopic analyses of BGO provide clear evidence for the incorporation of both boron and nitrogen into the graphene oxide framework. The FTIR spectrum displays a well-defined C-B vibrational band [40], and the XPS B1s region resolves into BC_3_ and BC_2_O configurations, confirming successful boron substitution [41]. Likewise, the N1s spectrum shows pyridinic, pyrrolic, and graphitic nitrogen, demonstrating that nitrogen is also chemically integrated into the lattice [39]. Although BGO and NGO were initially synthesized in the same autoclave, subsequent, independently prepared batches showed nitrogen in BGO, indicating that nitrogen incorporation is intrinsic to the hydrothermal conditions rather than an artifact of cross-contamination. Notably, prior studies have reported that low-level nitrogen incorporation can occur during modified Hummers or hydrothermal treatments, even in the absence of intentional nitrogen sources [42]. Together with literature showing that simultaneous B and N incorporation modulates charge distribution, enhances carrier density, and strengthens NIR absorption in graphene-based materials, the combined presence of both dopants is consistent with the superior photothermal response observed for BGO in this study [32,43].

The Raman spectra show the G band, confirming the presence of sp2-hybridized carbon atoms in GO, BGO, and NGO [44]. The appearance of the D and D′ bands indicate the presence of defects and disorder in the carbon structures. These bands arise from graphene domain edges, dopants, and other defects in the carbon network. The variation in the position of these bands among the samples suggests that the type and number of dopants significantly influence the structure and distribution of defects [44]. The presence of the 2D′ band in the D band and the D* and D″ bands in the G band underscores the complexity of doped structures. These additional bands indicate interaction between dopants and the carbon matrix, altering lattice vibrations and generating new features in the Raman spectrum [27]. Other authors have previously reported the appearance of the G* band in purified GO samples [44].

The D+D′ combination bands are activated in all samples regardless of defect concentration, indicating that this vibrational feature is an intrinsic consequence of the oxidized graphene framework and persists after heteroatom doping [44]. The ID/IG ratio of 1.55 for GO, 1.73 for BGO, and 1.54 for NGO, all below 3.5, indicates the high crystallinity of the samples [44]. This quantitative measure of defects in the carbon structure is crucial for assessing material quality. The variation in the D and D′ band positions across the samples directly reflects the influence of nitrogen or boron incorporation in the defect landscape and lattice distortion of graphene oxide [27].

Cytotoxicity assays show similar IC50 values for GO and NGO: 1025.26 µg/mL and 1319.81 µg/mL, respectively. Conversely, BGO exhibited the lowest toxicity, with an IC50 of 2695.03 µg/mL. These concentrations demonstrate the low toxicity of the studied materials, making them suitable for biomedical applications as proposed in this study. The low cytotoxicity of the samples is attributed to the excellent biocompatibility of carbon-based materials with various cells [45,46]. Furthermore, it is demonstrated that doping does not impair the intrinsic low cytotoxicity of GO and, in the case of BGO, even enhances it.

Anticancer studies performed in the T-47D breast cancer cell line demonstrate that light irradiation alone does not eliminate tumor cells. Similarly, the presence of the analyzed materials alone allows for the survival of 55.63%, 61.99%, and 49.27% of tumor cells for GO (1000 µg/mL), BGO (2000 µg/mL), and NGO (1000 µg/mL), respectively, indicating they are not effective on their own for eliminating cancer cells. However, all three samples show nearly complete elimination of tumor cells after 30 min of irradiation, leaving only 2.07%, 1.46%, and 2.017% of tumor cells for GO (1000 µg/mL), BGO (2000 µg/mL), and NGO (1000 µg/mL), respectively. This demonstrates excellent anti-tumor action under the photothermal effect, a result consistent with other studies [20,21,22,29,45].

The threshold temperatures of GO, BGO, and NGO exceed 42 °C (44.87 °C for GO, 55.91 °C for BGO, and 48.36 °C for NGO), which is the temperature at which thermal damage to cells begins. However, in all three cases, some cancer cells persist after treatment because the temperatures reached are below 60 °C, the temperature for irreversible cell death due to hyperthermia.

Furthermore, analysis of the samples’ relaxation times indicates similar anticancer activity in all three cases. NGO has the shortest relaxation time among the samples (2.21 min), suggesting it reaches its threshold temperature more quickly, thereby reducing the required irradiation times. BGO is the second material with a shorter relaxation time (2.53 min), indicating shorter irradiation times. In contrast, GO has the longest relaxation time (4.35 min), implying it requires a longer irradiation time.

BGO achieves the highest temperature, exhibits lower cytotoxicity, and reaches its threshold temperature quickly, making it the most efficient for PTT applications. Additionally, functionalizing graphene oxide with nitrogen has shown significant improvement in its photothermal properties, as also proposed by various authors [20,21,22,29,45].

Based on atomistic simulations, the absorbance coefficient shows a main band around 290 nm, regardless of the type and percentage of doping in graphene. However, it is notable that both nitrogen and boron doping result in a decrease in the absorbance coefficient with increasing doping percentage. Additionally, in the region around 1425 nm, the absorbance coefficient increases with increasing doping percentage. This phenomenon is consistently present regardless of the type of doping used.

Nitrogen incorporation increases the Fermi level because each substitution introduces an additional electron into the π-system, reducing the density of accessible conduction-band states and lowering the absorbance intensity near 290 nm [47]. In contrast, boron substitution decreases the Fermi level by generating electron vacancies, which require higher-energy electronic transitions and similarly reduce absorbance in this region. Both dopants, therefore, modify the optical response through well-defined electronic mechanisms associated with donor or acceptor behavior [39].

It is important to emphasize that the DFT calculations were intentionally performed on pristine graphene lattice doped with nitrogen or boron, rather than on graphene oxide. This approach follows established computational practice, as fully modeling GO requires explicitly accounting for the random, non-periodic distribution of epoxide, hydroxyl, carbonyl, and carboxyl groups, which break lattice symmetry and demand supercells containing several hundred to several thousand atoms. Such systems are incompatible with high-resolution calculations of band structure, PDOS, and optical absorption using dense k-point and q-point meshes. Therefore, the oxygen-free graphene model is not intended to reproduce the absolute electronic or optical properties of GO, NGO, or BGO but rather to isolate and quantify the intrinsic electronic effects of substitutional N and B doping. This allows a mechanistic interpretation of experimentally observed trends, such as Fermi-level shifts, changes in the electronic density of states, and enhanced absorption upon doping, while acknowledging that the magnitude of these effects differs across oxygen-functionalized materials.

Furthermore, the absorbance enhancement near 1425 nm arises from intra-band transitions facilitated by the modified electronic structure of the doped systems. In nitrogen-doped graphene, the elevated Fermi level promotes low-energy transitions within the conduction band, whereas the hole-rich valence band of boron-doped graphene enables similarly low-energy transitions toward these states [39].

DFT calculations typically underestimate transition energies owing to the approximations intrinsic to exchange-correlation functionals and pseudopotentials [47]. This well-known effect shifts the predicted absorbance features toward longer wavelengths, consistent with the redshift observed at 1425 nm and with the superior photothermal performance of BGO under 635 nm irradiation.

Because the graphene-based model lacks oxygen functionalities, the theoretical results should be interpreted exclusively as qualitative descriptors of doping-induced electronic modifications. These calculations establish the direction and nature of the changes introduced by N and B dopants, rather than providing a direct quantitative match to the experimental GO-based systems. Future work incorporating explicit oxygen distributions or using large-scale ab initio or reactive molecular dynamics models would enable a more complete representation of GO, NGO, and BGO. However, the present simplified model remains appropriate for elucidating the fundamental electronic contributions of the dopants.

## 4. Materials and Methods

### 4.1. Synthesis of Graphene Oxide and Doping with Nitrogen and Boron

Graphene oxide was synthesized using the modified Hummers method. Initially, 2 g of graphite (Graflake, 99%, ≤150 μm, Nacional de Graphite, São Paulo, Brazil) and 120 mL of concentrated sulfuric acid (H_2_SO_4_, 99%, Sigma-Aldrich, Darmstadt, Germany) were placed in a 1000 mL two-neck round-bottom flask while stirring on an ice bath (temperature ≤ 15 °C) for 15 min. After this time, 7 g of potassium permanganate (KMnO_4_, ≥99%, Sigma-Aldrich, Darmstadt, Germany) was slowly added over approximately 15 min in the same ice bath. The solution was removed from the ice bath and stirred for 24 h. Subsequently, the solution was returned to the ice bath, and 200 mL of deionized water was added slowly. Then, 20 mL of hydrogen peroxide (H_2_O_2_, 35%, CRQ Química, São Paulo, Brazil) was added to eliminate excess oxidant. The dispersion was left to stand for 24 h and then filtered and washed with different solvents: 500 mL of deionized water, 250 mL of a 10% aqueous solution of hydrochloric acid, and 250 mL of ethanol (Sigma-Aldrich, Darmstadt, Germany). Finally, it was washed with deionized water several times until the solution pH exceeded 6, ensuring the removal of acidic residues.

For the exfoliation process, the aqueous graphene oxide solution obtained was placed in an ultrasonic bath (Elmasonic P, Elma Schmidbauer, Singen, Germany) and sonicated at 37 kHz with 100% power at a temperature below 15 °C for 1 h. The resulting sample was denoted as GO.

The functionalization of GO with boron and nitrogen was achieved via a simple hydrothermal reaction. For boron functionalization, 4.9 mL of the GO solution was mixed with 50 mL of a 1 M solution of boric acid (H_3_BO_3_, 99%, Sigma-Aldrich, Darmstadt, Germany). The resulting mixture was placed in a stainless-steel Teflon-lined autoclave at 120 °C and pressure for 2 h. The obtained sample, designated as BGO, was collected, washed with deionized water, and then filtered.

For nitrogen functionalization, 70 mL of the GO solution was mixed with 30 mL of ammonia (NH_3_, ≥99%, Sigma-Aldrich, Darmstadt, Germany). The resulting mixture was placed in a stainless-steel Teflon-lined autoclave at 80 °C and pressure for 3 h. The obtained NGO sample was collected, washed with deionized water, and filtered.

### 4.2. Characterization of GO, BGO, NGO

The samples obtained were characterized by FTIR spectroscopy using a FT/IR spectrometer (JASCO, Easton, MD, USA) at a resolution of 0.7 cm^−1^, covering wavelengths from 7800 to 350 cm^−1^. Additionally, Raman spectroscopy (HORIBA, Essex, UK) was employed excitation at 2.33 eV (532 nm). XPS analysis was performed on an ESCALAB 250Xi spectrometer (Thermo Fisher Scientific, Waltham, MA, USA) equipped with a hemispherical electron energy analyzer. XPS spectra were acquired using monochromatized Al Kα excitation energy (hν = 1486.7 eV). SEM micrographs were obtained using a Tescan Mira 3 microscope (Tescan, Brno, Czech Republic) with a Schottky field emitter. AFM imaging was performed on a Dimension Icon (Bruker, Portland, OR, USA) operated in tapping (AC) mode with a closed-loop scanner. Silicon cantilevers TESPA-V2 was used (Bruker, Portland, OR, USA), with nominal spring constant 37–44 N·m^−1^, resonance frequency 320–410 kHz, and nominal tip radius < 10 nm. The scan size was 2.0 μm, with an image resolution of 1024 × 1024 pixels, a line rate of 0.8 Hz, and a setpoint amplitude of 70% of the free oscillation amplitude. Samples were prepared by drop-casting 10 μL of an aqueous dispersion (100 μg·mL^−1^) onto SiO_2_/Si (300 nm), rinsing with deionized water, and drying under N_2_. Measurements were performed at 23 °C and 45% relative humidity inside an acoustic/anti-vibration enclosure.

Vertical calibration of the AFM scanner was verified prior to data acquisition using a certified step-height calibration grating model VGRP-20 (Bruker, Portland, OR, USA), with a nominal step height of 20 ± 0.6 nm. The calibration grating is ISO-traceable and was validated within the manufacturer’s recommended calibration interval. Raw AFM images were processed in Gwyddion 2.69 (SourceForge, Brno, Czech Republic) using first-order plane leveling and line-by-line baseline subtraction. No second-order curvature corrections or smoothing filters were applied for quantitative height analysis; a 3 × 3 median filter was used only for visual display. Thickness values were extracted from cross-section profiles across sheet edges, and *n* = 5 independent regions were measured for each sample

### 4.3. Cytotoxicity, Anticancer, and Photothermal Studies

Rhesus monkey kidney epithelial cells (LLC-MK2), provided by the Center for Research on Health in Latin America (CISeAL), Quito, Ecuador, were used for the cytotoxicity study. The cells were cultured for 24 h in modified Eagle’s medium (DMEM, Gibco, Thermo Fisher, Waltham, MA, USA) supplemented with 1% penicillin-streptomycin (Sigma Aldrich, Darmstadt, Germany), 1% sodium pyruvate (Sigma Aldrich, Darmstadt, Germany), and 20% fetal bovine serum (FBS, Sigma Aldrich, Darmstadt, Germany).

For the assay, 20,000 cells per well were seeded in a 96-well plate, with each well containing 100 μL of DMEM. The cells were incubated for 24 h in a Series II Water Jacket incubator (CO_2_ Incubator, Thermo Fisher, Waltham, MA, USA) at 37 °C with a 5% CO_2_ atmosphere and 98% relative humidity. Serial 1/2 dilutions of the test agents (GO, NGO, and BGO) were prepared, ranging from 4000 μg/mL to 3.91 μg/mL. To assess cell viability, 10 μL of sodium resazurin salt (3 mM) (Sigma Aldrich, Darmstadt, Germany) was added to each well and incubated under the same conditions. After 24 h, fluorescence was measured using excitation wavelengths of 530–560 nm and emission at 590 nm in a GloMax microplate reader (Promega Corporation, Madison, WI, USA). The data points obtained in the assay were fitted with a dose–response curve.

T-47D breast cancer cells provided by CISeAL (Quito, Ecuador) were used for anticancer assays under photothermal conditions. These cells were cultured under the same conditions mentioned in the cytotoxicity section. Based on the cytotoxicity results, concentrations below the IC_50_ obtained in the assay were selected: 1000 μg/mL for GO and NGO and 2000 μg/mL for BGO.

For the assay, 1 mL of a homogeneous mixture of the test agents at the concentrations mentioned above, along with cells (2 × 10^5^), was placed in plastic wells. These mixtures were irradiated for 30 min using an LED lamp with a wavelength of 635 nm and a power of 65.5 mW/cm^2^, covered with aluminum foil. The wells containing the samples were kept at room temperature before the assay began. These assays were conducted with their respective controls, consisting of cell culture medium with cells, and all treatments were kept in the dark. After 48 h, fluorescence was measured using excitation wavelengths of 530–560 nm and emission at 590 nm in a GloMax microplate reader (Promega Corporation, Madison, WI, USA). The data obtained were subjected to analysis of variance (ANOVA) at the *p* ≤ 0.001 significance level.

To evaluate the photothermal effects of GO, NGO, and BGO, 1 mL of each sample solution at the concentrations listed above was placed in plastic wells. These samples were irradiated under the conditions described in the previous anticancer assay. Upon irradiation onset, the samples’ temperature was measured every 3 min using a digital thermometer (SCANMED, Warsaw, Poland sensitivity of 0.1 °C) until a threshold temperature was reached and stabilized. The obtained data were fitted with a curve as follows:
(1)
$$T=T_{max}-\left(T_{max}-T_{0}\right)e^{-\frac{t}{\tau }}$$

where 
$T_{max}$
 is the threshold temperature reached by the sample due to irradiation, 
$T_{0}$
 is the initial temperature of the sample, *t* is the time elapsed since the start of irradiation, and *τ* is the relaxation time of the sample (the characteristic time it takes for the sample temperature to stabilize).

### 4.4. Atomistic Simulations

Atomistic simulations were performed using ab initio DFT [31,48], implemented in the SIESTA code [28]. An idealized pristine graphene supercell doped with nitrogen or boron was employed, a widely used approach to isolate the intrinsic electronic effects of substitutional dopants. Explicitly modeling graphene oxide with its heterogeneous and non-periodic distribution of epoxide, hydroxyl, carbonyl, and carboxyl groups would require non-tractable supercells containing hundreds to thousands of atoms, preventing accurate band structure, PDOS, and optical calculations. The simplified graphene-based model, therefore, provides mechanistic insight into the electronic modifications induced by N and B dopants, rather than a direct reproduction of the properties of GO, NGO, or BGO.

Troullier-Martins norm-conserving pseudopotentials [34] were employed with an energy cutoff of 500 Ry. Atomic orbitals were based on a DZP basis set, with a basis-set confinement energy cutoff of 0.02 Ry. The PBEsol exchange-correlation functional [49] was used, and Brillouin zone sampling was performed using a Monkhorst-Pack grid of 4 × 4 × 1 k-points [50]. All geometries were optimized with forces less than 0.01 eV/Å. A vacuum spacing of at least 20 Å along the z-direction was used to avoid interaction between periodic images. For PDOS calculations, a 4 × 4 × 1 k-point mesh in the Brillouin zone was used. Optical properties were computed using a 50 × 50 × 1 q-point optical mesh and a Gaussian broadening of 0.1 eV.

The simulations employed an orthorhombic supercell containing 60 carbon atoms. To simulate doping, one carbon atom was progressively replaced with a dopant atom in each simulation, as illustrated in Figure 12.

## 5. Conclusions

Based on spectroscopic tests and morphological analyses, it has been confirmed that the synthesis methodology produced GO and its NGO and BGO variants. FTIR and XPS spectra revealed functional groups associated with carbon oxidation, while SEM and AFM micrographs demonstrated the characteristic graphene oxide sheet structure.

Percentage analysis of the XPS spectra showed no significant loss of oxygen functional groups during functionalization. Furthermore, the incorporation of nitrogen and boron into the doped graphene networks was confirmed, indicating successful doping.

Additionally, Raman spectra revealed defects in the carbon network, primarily due to oxygen functional groups and, to a lesser extent, to doping in the samples. However, the ratio of the D and G bands intensities suggests that the samples maintain high crystallinity. This indicates that the defects do not significantly compromise the structural quality of the material.

Cytotoxicity assays demonstrated that GO, NGO, and BGO exhibit low toxicity, with BGO being the least toxic, making them suitable for biomedical applications. Furthermore, anticancer studies demonstrated that all three samples exhibited excellent antitumor activity under photothermal irradiation, with BGO standing out for its ability to reach threshold temperatures and for faster relaxation times.

Atomistic simulations revealed that the absorbance coefficient exhibits a main band around 290 nm, decreasing exponentially with increasing doping percentage and increasing exponentially in the region around 1425 nm. Changes in the Fermi energy and electron density of doped graphene explain these phenomena.

It is important to note that DFT-based simulations tend to underestimate the energy gap, leading to a redshift in the absorbance bands. Therefore, the increase in the absorbance coefficient around 1425 nm explains NGO’s enhanced photothermal transformation efficiency compared to GO and the notable efficiency in the photothermal transformation of BGO under red light.

In conclusion, the functionalization of graphene oxide with nitrogen and boron successfully enhanced its photothermal properties, positioning BGO as the most efficient of the three studied materials for PTT applications. The low cytotoxicity and high photothermal efficiency of these materials open new opportunities for their use in advanced biomedical applications.

## Figures and Tables

**Figure 1 ijms-26-11771-f001:**
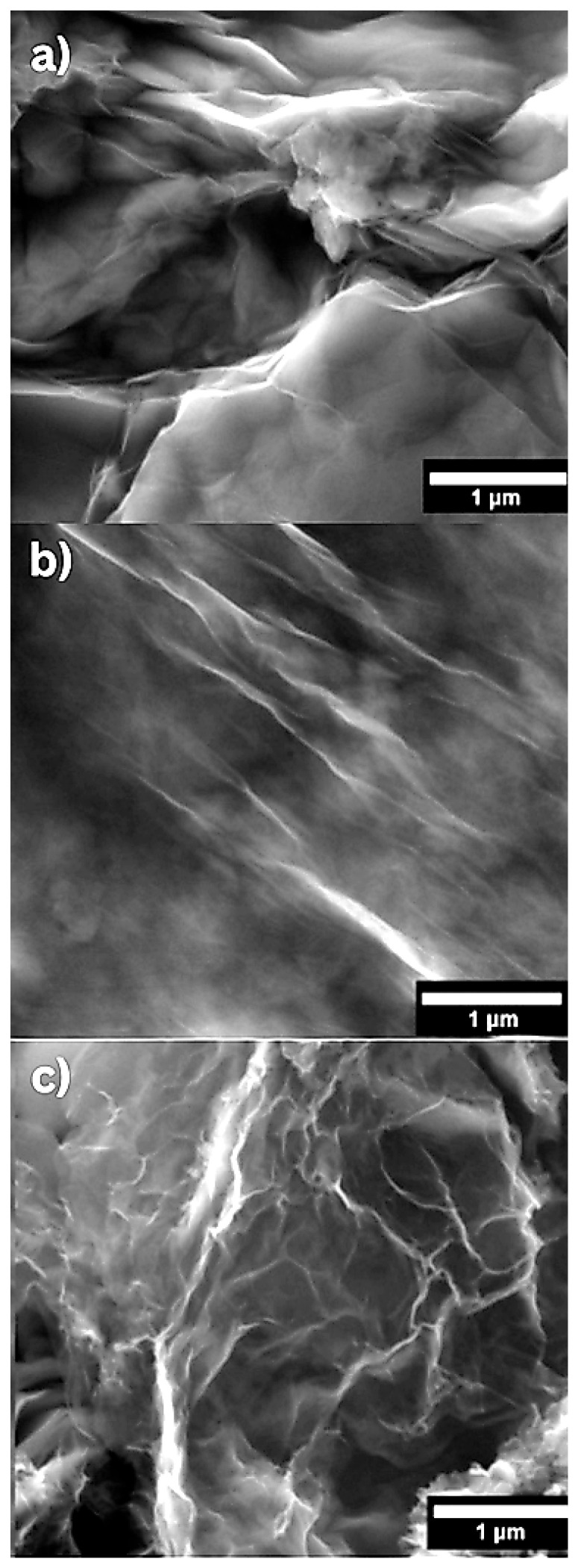
SEM micrographs of graphene oxide and doped graphene oxide: (**a**) graphene oxide, (**b**) boron-doped graphene oxide, and (**c**) nitrogen-doped graphene oxide.

**Figure 2 ijms-26-11771-f002:**
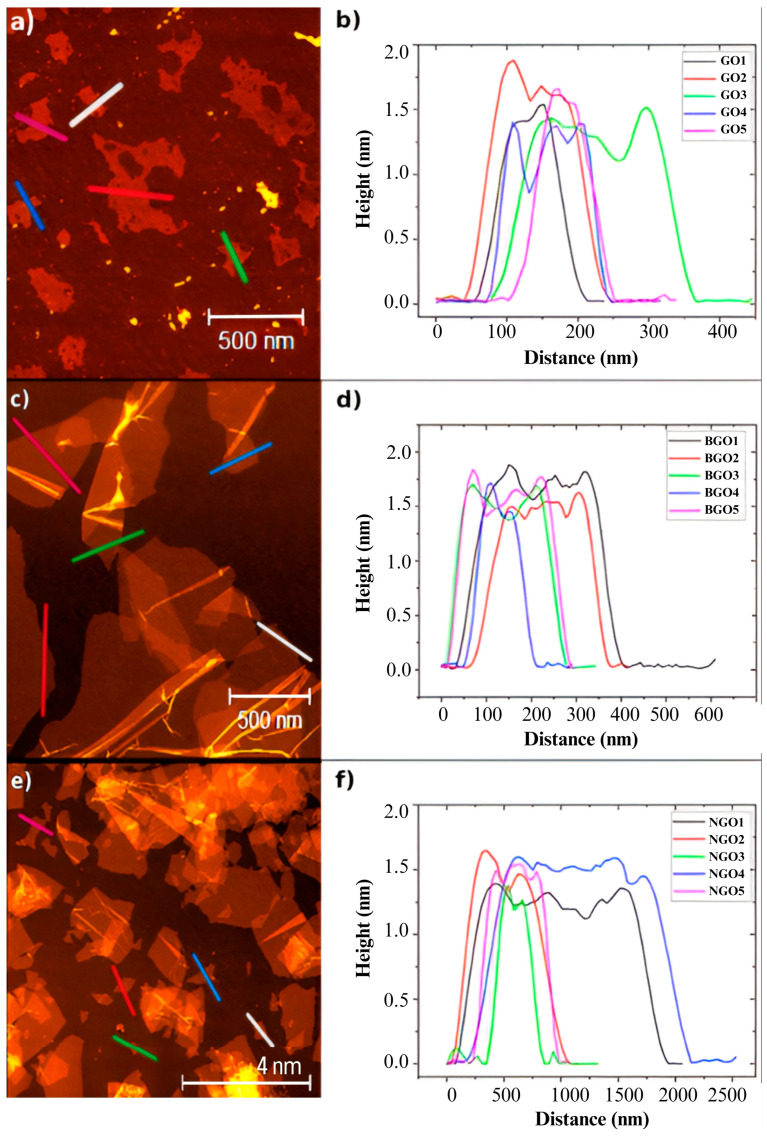
Thickness characterization (each line marked in the micrograph represents the areas in which a height profile was generated): (**a**) AFM graphene oxide micrograph, (**b**) Graphene oxide height profile, (**c**) AFM BGO micrograph, (**d**) BGO height profile, (**e**) AFM NGO micrograph, and (**f**) NGO height profile.

**Figure 3 ijms-26-11771-f003:**
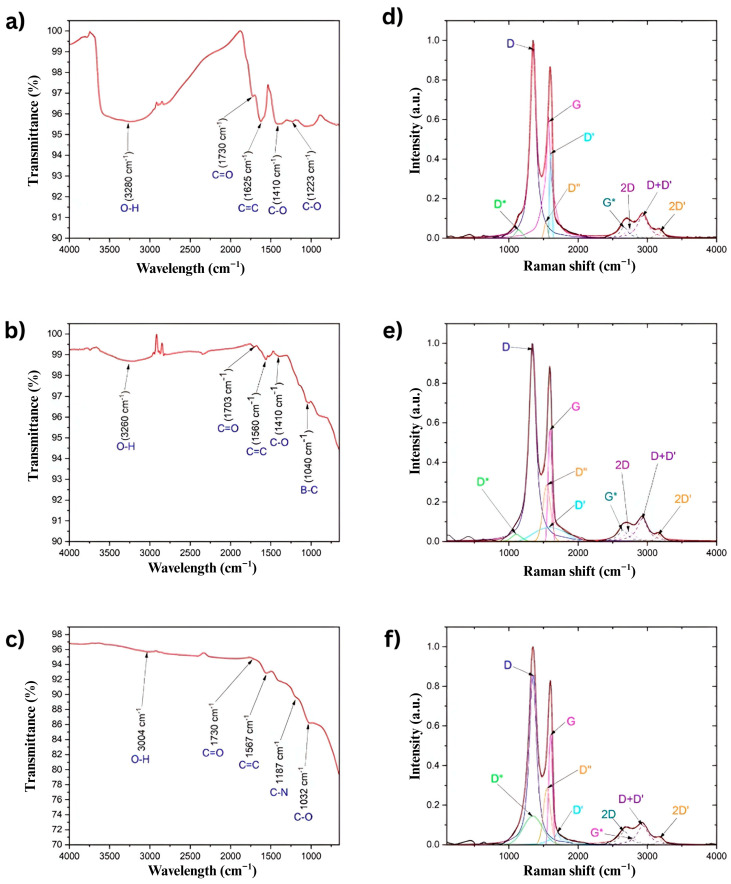
FTIR spectra of graphene oxide and doped graphene oxide: (**a**) GO, (**b**) BGO, (**c**) NGO. Raman spectra of graphene oxide and doped graphene oxide: (**d**) GO, (**e**) BGO, and (**f**) NGO.

**Figure 4 ijms-26-11771-f004:**
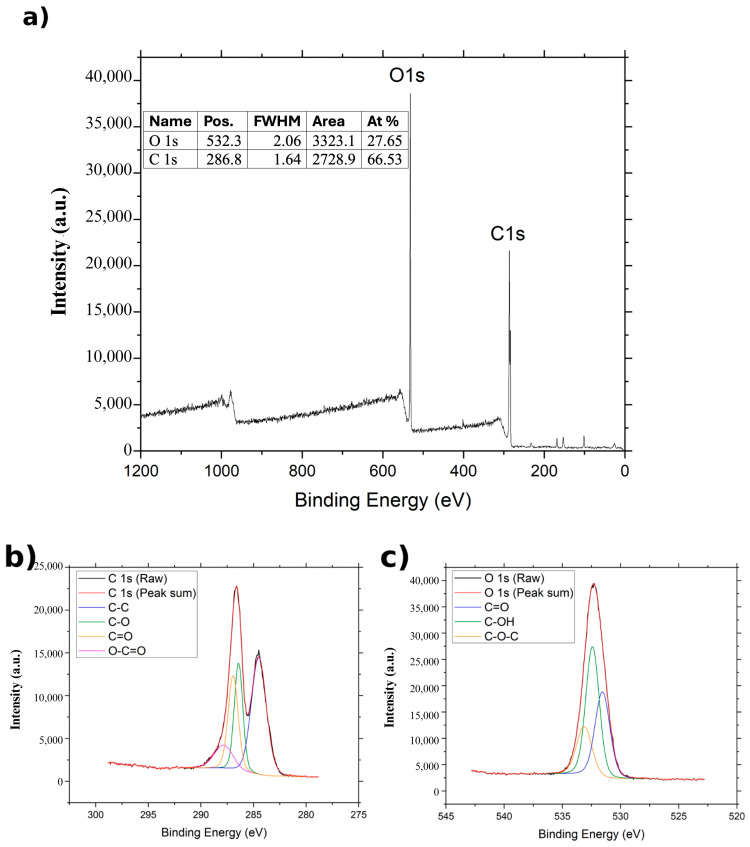
XPS spectrum of GO: (**a**) XPS spectrum of GO, (**b**) XPS spectrum of C1s region of GO, and (**c**) XPS spectrum of O1s region of GO.

**Figure 5 ijms-26-11771-f005:**
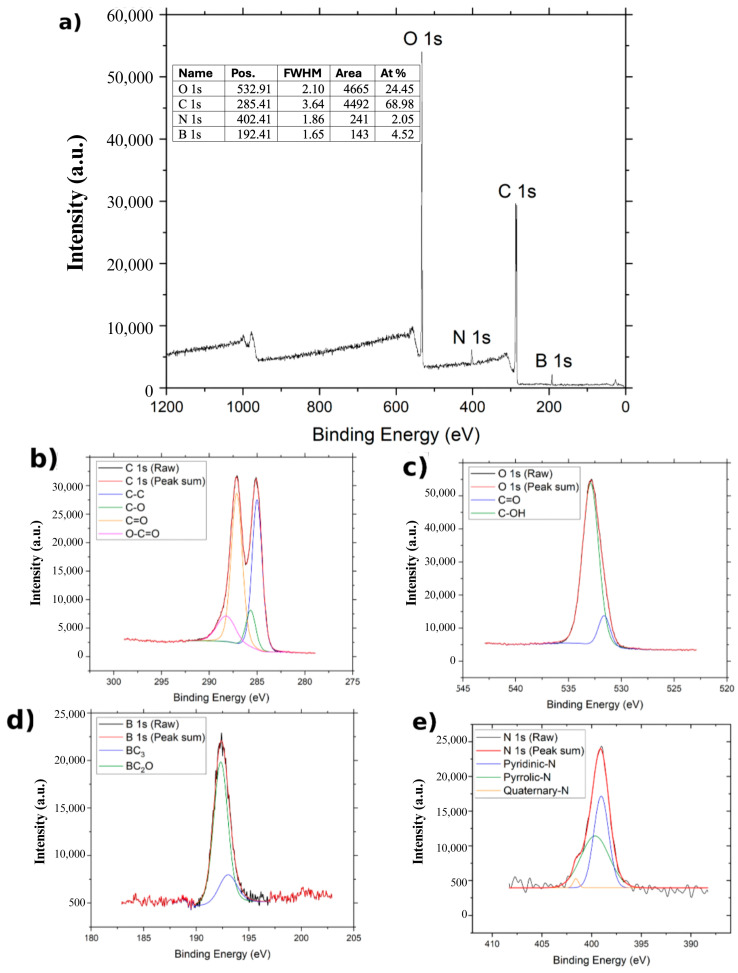
XPS spectrum of BGO: (**a**) XPS spectrum of BGO, (**b**) XPS spectrum of C1s region of BGO, (**c**) XPS spectrum of O1s region of GO, (**d**) XPS spectrum of B1s region of BGO and (**e**) XPS spectrum of N1s region of BGO.

**Figure 6 ijms-26-11771-f006:**
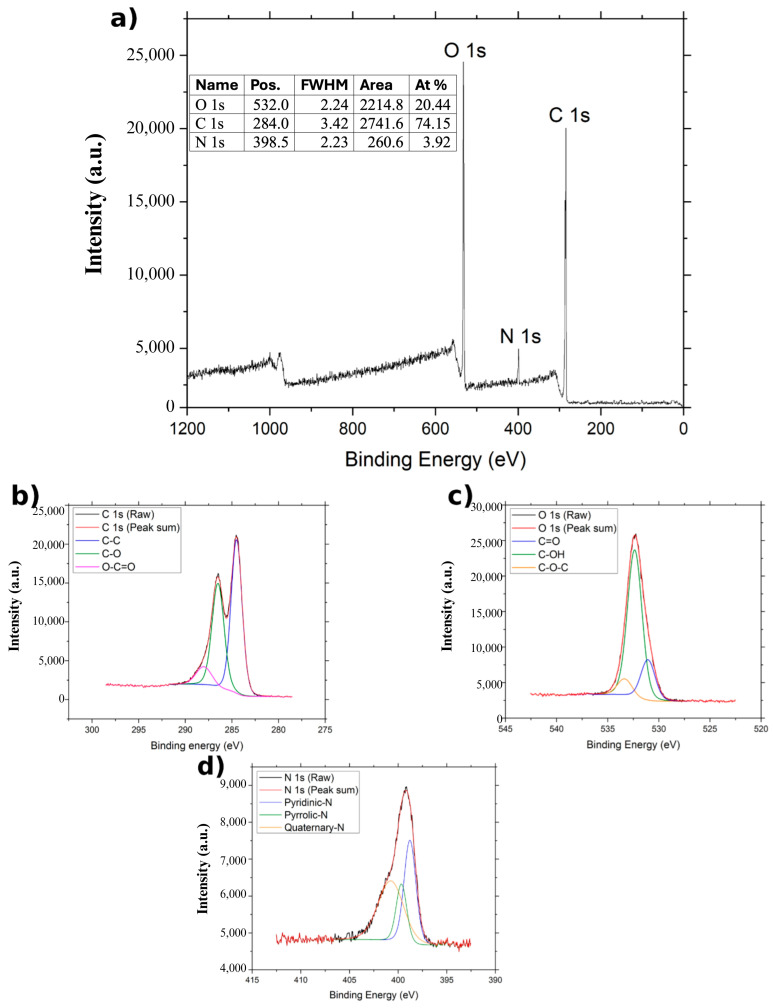
XPS spectrum of NGO: (**a**) XPS spectrum of NGO, (**b**) XPS spectrum of C1s region of NGO, (**c**) XPS spectrum of O1s region of NGO, and (**d**) XPS spectrum of N1s region of NGO.

**Figure 7 ijms-26-11771-f007:**
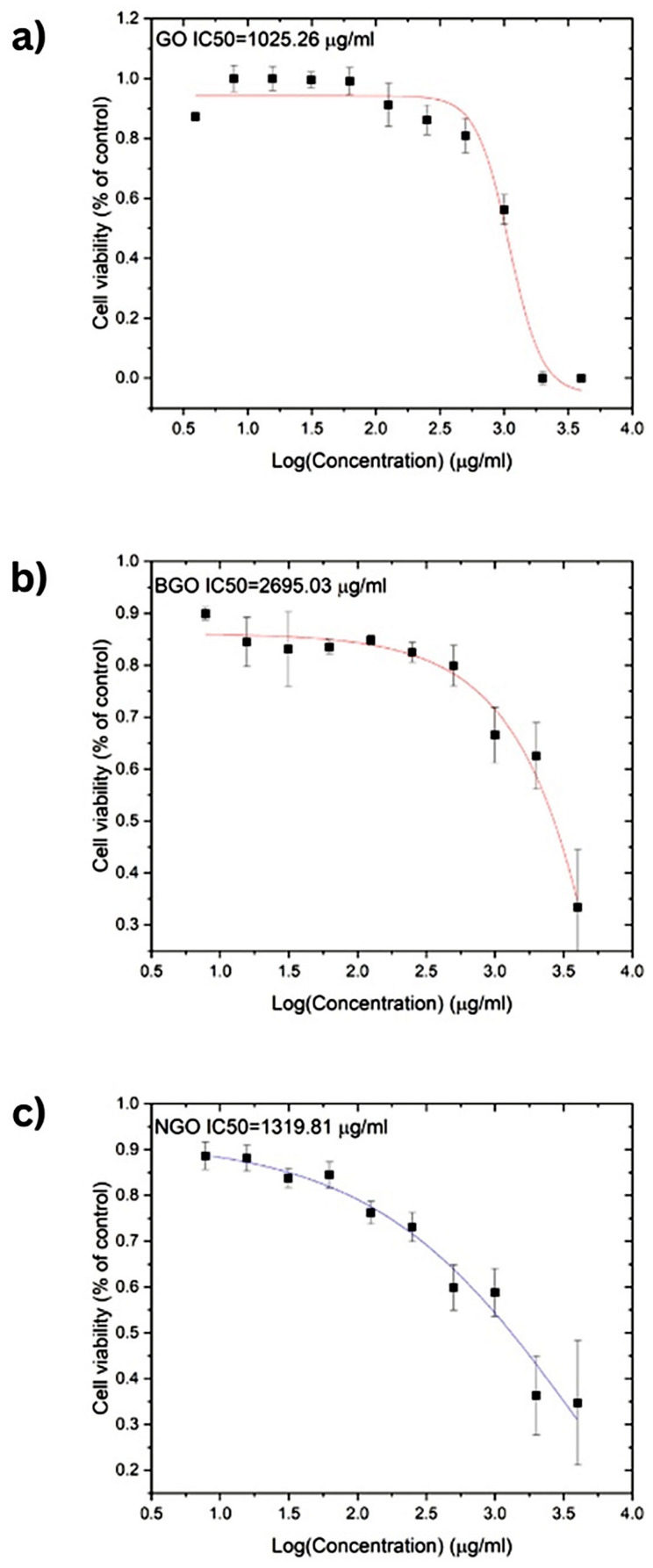
Cytotoxicity assay results and corresponding dose–response curve fits for (**a**) GO (fitting curve in red), (**b**) BGO (fitting curve in red), and (**c**) NGO (fitting curve in blue) in T-47D breast cancer cells. (IC_50_ values were calculated based on fluorescence viability data using resazurin dye after 24 h of treatment. Error bars represent standard deviations (*n* = 3).

**Figure 8 ijms-26-11771-f008:**
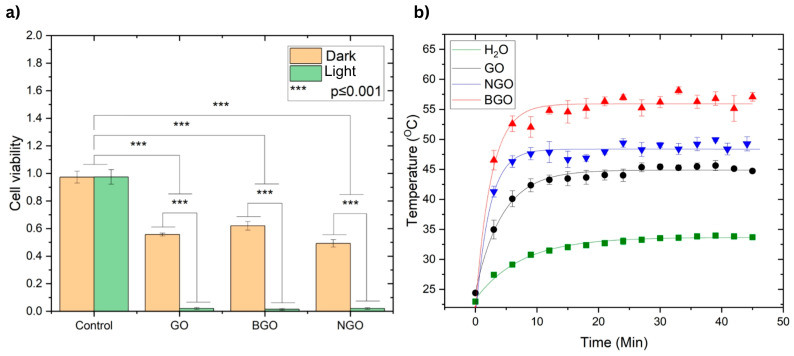
(**a**) Anticancer effect based on PTT for tumor cells with the T-47D cell line (breast cancer). The concentrations of GO and NGO were 1000 µg/mL, and for BGO, 2000 µg/mL. The samples were irradiated for 30 min. Significant differences were found according to Tukey’s test (*p* ≤ 0.001). (**b**) Photothermal heating curves: H_2_O (green square), GO (black circles), NGO (blue inverted triangles), and BGO (red triangles). The experiments were performed with a concentration of 1000 µg/mL for GO and NGO and 2000 µg/mL for BGO. Error bars represent standard deviations (*n* = 3).

**Figure 9 ijms-26-11771-f009:**
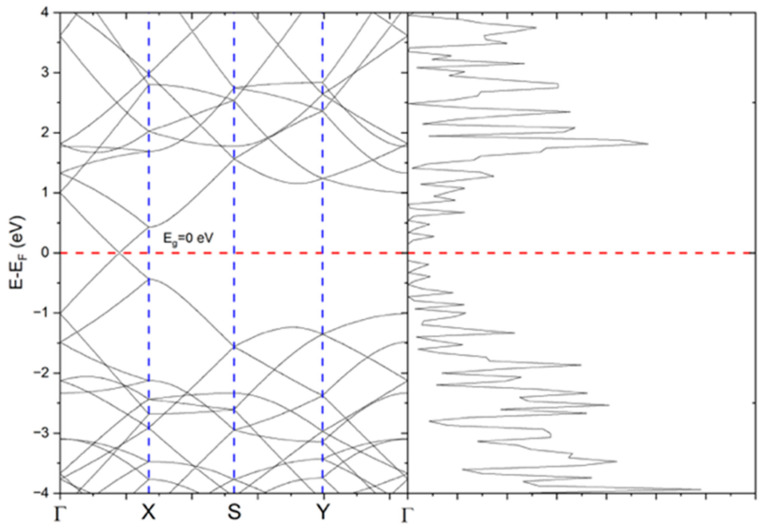
Band structure and density of states for a graphene lattice composed of 60 atoms organized in an orthorhombic superlattice. The red dashed line marks the Fermi level (E − E_F = 0 eV). The blue dashed vertical lines indicate the high-symmetry points along the k-path (Γ–X–S–Y–Γ) used for the band-structure calculation.

**Figure 10 ijms-26-11771-f010:**
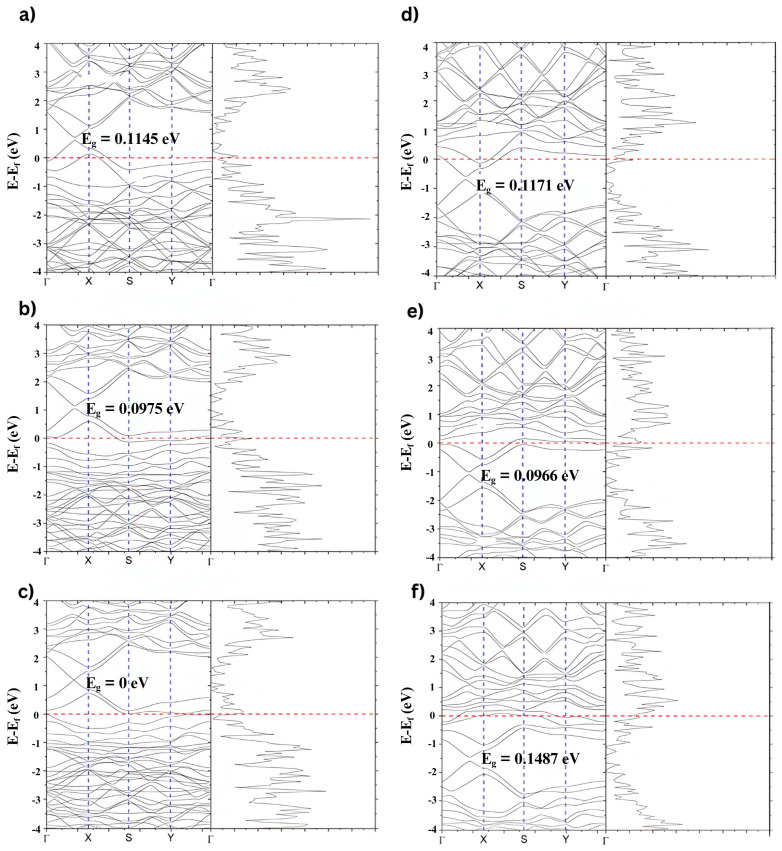
Band structure and density of states of a graphene lattice doped at different concentrations, organized in an orthorhombic superlattice. (**a**) Graphene doped with 1.67% boron, (**b**) Graphene doped with 5% boron, (**c**) Graphene doped with 6.67% boron, (**d**) Graphene doped with 1.67% nitrogen, (**e**) Graphene doped with 5% nitrogen, and (**f**) Graphene doped with 10% nitrogen. The red dashed line marks the Fermi level (E − E_F = 0 eV). The blue dashed vertical lines indicate the high-symmetry points along the k-path (Γ–X–S–Y–Γ) used for the band-structure calculation.

**Figure 11 ijms-26-11771-f011:**
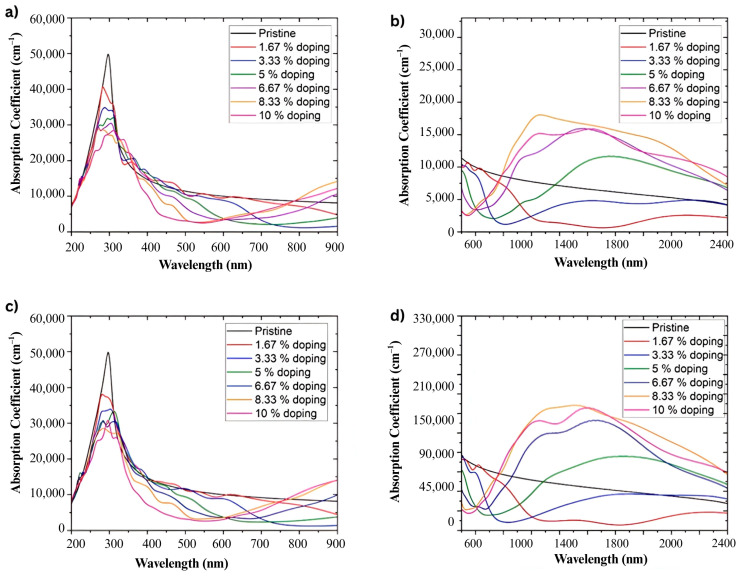
Band structure and density of states of a graphene lattice doped at different concentrations, organized in an orthorhombic superlattice. (**a**) Graphene doped with 1.67% boron, (**b**) Graphene doped with 3.33% boron, (**c**) Graphene doped with 5% boron, and (**d**) Graphene doped with 6.67% boron.

**Figure 12 ijms-26-11771-f012:**
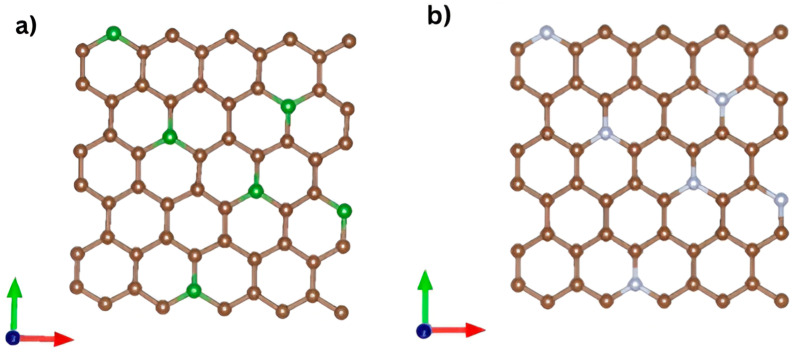
Models used in the simulations: (**a**) Graphene lattice doped with 10% nitrogen, where brown and blue spheres represent C and N atoms, respectively. (**b**) Graphene lattice doped with 10% boron, where brown and green spheres represent C and B atoms, respectively.

## Data Availability

The original contributions presented in this study are included in the article. Further inquiries can be directed to the corresponding author.

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
