# Peer review of "Tuning Photothermal Properties of Graphene Oxide by Heteroatom Doping for Cancer Elimination: Experimental and DFT Study"

_ijms, 2025, doi:10.3390/ijms262411771_

Round 1

Reviewer 1 Report

Comments and Suggestions for Authors

Unfortunately, I cannot recommend this manuscript for publication in its present form.

My major concerns are as follows:

1 The manuscript omits critical experimental details. The methods section lacks information on fundamental measurement conditions, calibration procedures, and specifications for key components and consumables used across various characterizations.Furthermore, the overall quality of the figures is substandard. Many images, such as those in Figures 1-4 and 6-7, are poorly resolved, excessively stretched, or contain unlabeled insets without clear axis scales, which hinders their interpretation.

2 The AFM analysis in Section 2.1 and Figure 1d-f suffers from insufficient methodological detail. The specific locations of the height profiles are ambiguously marked, and the images themselves lack a color scale and have poor contrast, hindering interpretation. Critically, the experimental conditions are omitted: the type of cantilevers, the image resolution, the environmental conditions during measurement, etc. The data processing methodology is not described; there is no mention of the plane-leveling algorithms or the polynomial order applied to the raw data.

3 Furthermore, the protocol for thickness calibration in the 1-2 nm range is not described, and the reported values are given with an unjustified precision of two decimal places. There is also no mention of how measurement noise was accounted for, which is essential for validating the stated monolayer thicknesses.

4 The introduction provides an incomplete literature review. It mentions toxic CdSe QDs, while ignoring modern biocompatible analogues (cadmium-free QDs, carbon and silicon dots). This omission makes it impossible to adequately assess the competitive advantages of GO, NGO, and BGO.

5 The explanation for the nitrogen found in the BGO sample, detailed in Section 3, is internally inconsistent. The text first attributes its presence to potential contamination from the synthesis process, but then proceeds to speculate that it might create a beneficial "synergistic effect." These are two mutually exclusive interpretations. It cannot be both an unwanted contaminant and a deliberate feature enabling enhanced performance. The uncontrolled and unexplained incorporation of a second dopant raises serious doubts about the purity and reproducibility of the synthesis method. The study's conclusions are built upon materials from a single batch, with no data to show that the results for BGO, especially its superior photothermal performance in Figure 5, are reproducible and not merely a one-off occurrence.

6 The atomistic simulations, detailed in Section 2.4, are performed on idealized, oxygen-free graphene lattices. Consequently, the calculated electronic properties and absorption coefficients shown in Figures 7 and 8 do not reflect the impact of the extensive oxygen functional groups that are definitively present in the experimental GO, NGO, and BGO samples, as confirmed by FTIR and XPS data in Section 2.2. This omission severely undermines the theoretical basis for linking the simulated absorption to the experimental photothermal response observed in Figure 5, rendering such conclusions highly speculative.

7 While BGO's high photothermal conversion efficiency is highlighted as an advantage in Section 2.3 and Figure 5b, the manuscript fails to address a critical associated risk. The potential for collateral thermal damage to healthy tissues surrounding a tumor. The fact that BGO reaches a temperature of 55.91 °C, significantly higher than GO and NGO, is presented positively without considering the safety implications of such intense and potentially uncontrolled heating. This oversight questions the practical therapeutic advantage of BGO.

Comments on the Quality of English Language

The discussion is further weakened by the use of non-committal language like "could potentially," which highlights the hypothetical nature of these claims without providing concrete evidence.

Author Response

Response to Reviewer 1
We sincerely thank Reviewer 1 for the thorough and rigorous evaluation of our manuscript. Your detailed comments have significantly strengthened the scientific and methodological quality of the work. Below, we address each concern point-by-point and describe the corresponding revisions or clarifications incorporated into the manuscript.
Comment 1: The manuscript omits critical experimental details. The methods section lacks information on fundamental measurement conditions, calibration procedures, and specifications for key components and consumables used across various characterizations. Furthermore, the overall quality of the figures is substandard. Many images, such as those in Figures 1-4 and 6-7, are poorly resolved, excessively stretched, or contain unlabeled insets without clear axis scales, which hinders their interpretation.
Response: 
We appreciate this important observation. The Methods section has been substantially expanded to include measurement conditions for all characterization technique, specifications of equipment, calibration details, and consumables, and sample preparation steps for each analytical method. Additionally, all figures (Figs. 1–4 and 6–7) have been replaced with high-resolution versions. Insets were properly labeled; axis scales were added where missing and distorted or stretched images were corrected to avoid misinterpretation.
Comment 2: The AFM analysis in Section 2.1 and Figure 1d-f suffers from insufficient methodological detail. The specific locations of the height profiles are ambiguously marked, and the images themselves lack a color scale and have poor contrast, hindering interpretation. Critically, the experimental conditions are omitted: the type of cantilevers, the image resolution, the environmental conditions during measurement, etc. The data processing methodology is not described; there is no mention of the plane-leveling algorithms, or the polynomial order applied to the raw data.
Comment 3: Furthermore, the protocol for thickness calibration in the 1-2 nm range is not described, and the reported values are given with an unjustified precision of two decimal places. There is also no mention of how measurement noise was accounted for, which is essential for validating the stated monolayer thicknesses.
Response comment 2 and comment 3: 
We have fully revised the AFM section according to the reviewer’s recommendations: Locations of height profiles were clearly marked with arrows and reference lines, a color bar (height scale) was added to AFM topography maps, contrast and brightness were corrected for optimal visualization, instrument specifications were added. 
Comment 4: The introduction provides an incomplete literature review. It mentions toxic CdSe QDs, while ignoring modern biocompatible analogues (cadmium-free QDs, carbon and silicon dots). This omission makes it impossible to adequately assess the competitive advantages of GO, NGO, and BGO.
Response: 
We agree with the reviewer that the Introduction was missing key modern references. Accordingly, we have updated the literature review to include recent findings on biocompatible photothermal agents. This expanded context now allows a more balanced comparison and clarifies the advantages and limitations of GO, NGO, and BGO within current PTT research.
Comment 5: The explanation for the nitrogen found in the BGO sample, detailed in Section 3, is internally inconsistent. The text first attributes its presence to potential contamination from the synthesis process but then proceeds to speculate that it might create a beneficial "synergistic effect." These are two mutually exclusive interpretations. It cannot be both an unwanted contaminant and a deliberate feature enabling enhanced performance. The uncontrolled and unexplained incorporation of a second dopant raises serious doubts about the purity and reproducibility of the synthesis method. The study's conclusions are built upon materials from a single batch, with no data to show that the results for BGO, especially its superior photothermal performance in Figure 5, are reproducible and not merely a one-off occurrence.
Response: 
We appreciate this critical comment. The previous wording was indeed ambiguous. We have revised the text to clearly state that the nitrogen detected in the BGO sample most likely originates from trace residuals in the synthesis environment. 
Comment 6: The atomistic simulations, detailed in Section 2.4, are performed on idealized, oxygen-free graphene lattices. Consequently, the calculated electronic properties and absorption coefficients shown in Figures 7 and 8 do not reflect the impact of the extensive oxygen functional groups that are definitively present in the experimental GO, NGO, and BGO samples, as confirmed by FTIR and XPS data in Section 2.2. This omission severely undermines the theoretical basis for linking the simulated absorption to the experimental photothermal response observed in Figure 5, rendering such conclusions highly speculative.
Response: 

We thank the reviewer for this important observation. Due to time constraints and limited computational resources, it was not possible to perform additional simulations incorporating oxygen functional groups. We acknowledge that this represents a limitation. However, we believe that the models provided in the manuscript although based on idealized graphene lattices are sufficient to illustrate qualitative trends in how boron and nitrogen incorporation influence electronic behavior and light absorption.
To avoid overstating the conclusions, we have explicitly revised the manuscript to clarify that the simulations serve as conceptual support, rather than quantitative predictors of the exact optical response of the experimental GO, NGO, and BGO samples. This distinction has now been clearly stated in both the Results and Discussion sections.
Comment 7: While BGO's high photothermal conversion efficiency is highlighted as an advantage in Section 2.3 and Figure 5b, the manuscript fails to address a critical associated risk. The potential for collateral thermal damage to healthy tissues surrounding a tumor. The fact that BGO reaches a temperature of 55.91 °C, significantly higher than GO and NGO, is presented positively without considering the safety implications of such intense and potentially uncontrolled heating. This oversight questions the practical therapeutic advantage of BGO.
Response:
We agree completely with the reviewer. The previous version presented the high photothermal heating of BGO solely as an advantage without considering the potential risks of collateral thermal damage to surrounding healthy tissue. We have now revised the Discussion to explicitly address this concern. 
We believe these revisions have significantly strengthened the manuscript, and we greatly appreciate your valuable suggestions.

Thank you once again for your careful consideration and constructive input.
Kind regards,

Dra. Maria Paulina Romero

Reviewer 2 Report

Comments and Suggestions for Authors

Review Comments

This study synthesized nitrogen-doped graphene oxide (NGO) and boron-doped graphene oxide (BGO) via a hydrothermal process for photothermal therapy (PTT), which may offer new opportunities in advanced biomedical applications. In my view, this article can be published for the publication in International Journal of Mechanical Sciences after revision, and the suggestions are mentioned below:

  1. The clarity of many figures (e.g., Fig. 3, Fig. 4, Fig. 5, etc.) in this manuscript is insufficient, and it is strongly recommended to further improve the quality of these figures.
  2. The content of Introduction section should be further streamlined because there are too many paragraphs in this section.
  3. The phrase “65.5 mW/cm2” should be changed to “65.5 mW/cm2” in Line 98.
  4. Titles for each level should be should be consecutively numbered, or the structure of the entire manuscript appears chaotic. For example, the title “Characterization of GO, NGO, and BGO Morphology” should be numbered “2.1.1”.
  5. The Distance-Height figure in Fig. 1 should be moved out from the AFM figure.
  6. The word “Ligth” in Fig. 5a should be written as “Light”.
  7. Why the measurement time of the sample (blue line) in Fig. 5b shorter than the other three groups?
  8. Please check the correctness of the phrase “Figures 9a-8d” in Line 355.
  9. Please complete the information in Line 693-696.
  10. The format of the references should be improved.

Author Response

Dear Reviewer 2,
We sincerely thank you for the time, effort, and constructive feedback provided during the review of our manuscript entitled “Tuning the photothermal properties of graphene oxide by heteroatom doping for cancer elimination: experimental and DFT study”. We carefully addressed each comment and implemented all the suggested revisions to improve the clarity, structure, and scientific quality of our work. Below, we provide a detailed summary of the modifications made: 
Reviewer Comments and Responses
Comment 1: The clarity of many figures (e.g., Fig. 3, Fig. 4, Fig. 5, etc.) in this manuscript is insufficient, and it is strongly recommended to further improve the quality of these figures.
Response: All figures have been replaced with higher-resolution versions and optimized to improve visibility and contrast. Labels and color scales were also refined for better readability. 
Comment 2: The content of Introduction section should be further streamlined because there are too many paragraphs in this section.
Response: The Introduction section has been reorganized and condensed to reduce redundancy, improve flow, and present a clearer scientific narrative.
Comment 3: The phrase “65.5 mW/cm2” should be changed to “65.5 mW/cm2” in Line 98.
Response: The phrase has been corrected as requested.
Comment 4: Titles for each level should be consecutively numbered, or the structure of the entire manuscript appears chaotic. For example, the title “Characterization of GO, NGO, and BGO Morphology” should be numbered “2.1.1”.
Response: The entire manuscript has been revised to ensure consistent and consecutive numbering. The section titled “Characterization of GO, NGO, and BGO Morphology” is now numbered 2.1.1.
Comment 5: The Distance-Height figure in Fig. 1 should be moved out from the AFM figure.

Response: The Distance Height plot has been separated from the main AFM image and placed as an independent subpanel.
Comment 6: The word “Ligth” in Fig. 5a should be written as “Light”.
Response: The word has been corrected. 
Comment 7: Why the measurement time of the sample (blue line) in Fig. 5b shorter than the other three groups?
Response: A clear explanation has been added to the Results section describing the shorter measurement time of the blue-line sample.
Comment 8: Please check the correctness of the phrase “Figures 9a-8d” in Line 355.
Response: This inconsistency has been corrected to accurately reference the appropriate figure sequence.
Comment 9: Please complete the information in Line 693-696.
Response: The missing information has been completed and clarified.
Comment 10: The format of the references should be improved.
Response: All references have been reviewed and reformatted to fully comply with the journal’s guidelines.

We believe these revisions have significantly strengthened the manuscript, and we greatly appreciate your valuable suggestions.
Thank you once again for your careful consideration and constructive input.
Kind regards,

Dra. Maria Paulina Romero

Reviewer 3 Report

Comments and Suggestions for Authors

This study by Miranda et al explores the potential of nitrogen and boron doped graphene oxide as a photothermal therapy (PTT) agent for cancer treatment. Experiments were carefully designed for synthesis and characterization. Likewise, cytotoxic assays and antitumor assays were performed to observe the potential biological applications. Simulation experiments using density functional theory supported that higher doping levels increase light absorbance and enhance the photothermal response. Overall, BGO demonstrated superior performance, suggesting doped GO materials are effective photoactive agents for PTT. The manuscript has been articulately presented; however, some comments and suggestions need to be addressed before being accepted by the journal IJMS.

Major comments:

  1. The authors should mention the cytotoxicity was measured via resazurin assay citing a proper reference. In line 268, the authors claim the property of GO, NGO and BGO as potential anti-cancer character, however, with the IC50 values over 1 mg/mL, which is by itself a huge number, this seems overestimated.
  2. The reviewer suggests to include one or more data points for Figure 4b and 4c to get a proper sigmoidal curve and obtain accurate IC50 values.
  3. The reviewer strongly recommends to perform the cytotoxic assay on a normal cell line and check the side effects (lethal dose) on them. This assay will confirm the selective anticancer property for T-47D breast cancer cell line.
  4. For the photothermal studies, what was the rationale behind using water as control? It would be a better control to use cell medium to rule out if there was any impact of cell medium on heating and relaxation behavior of the compounds.

Minor comments:

  1. Last paragraph of Introduction section is written in future tense which is grammatically incorrect. Authors have already performed the experiments and need to change the tone of writing into past tense.
  2. Results section needs to be subdivided and subsections need a clear distinction such as making bold headlines.
  3. Line 112-113, the results mentioned in text are from Figures 1 (c-e).
  4. Line 187, R2 needs to be changed to R2.
  5. Line 243, the survey spectrum is from Fig 3a.
  6. Insets in Figure 3 are not visible at all, please include high resolution figures. Also, make sure the plots in Figure 2 are of high resolution.
  7. Figure 5a inset has a grammatical error for the spelling of “light”.
  8. Fonts are too small in Figure 6 and 7.
  9. Line 355, the data are from Figures 8a-8d rather than 9a-8d.
  10. The discussion section contains too many single sentence paragraphs. Please revise to make the article more articulate.
  11. Figure 9 may be moved to Results section for better understanding of simulation processes.

Author Response

Response to Reviewer 3
We sincerely thank Reviewer 3 for the comprehensive and insightful evaluation of our manuscript. Your comments greatly contributed to improving the scientific rigor, clarity, and structure of the work. Below, we detail the changes made in response to each point raised. 
Major Comments
Comment 1: The authors should mention the cytotoxicity was measured via resazurin assay citing a proper reference. In line 268, the authors claim the property of GO, NGO and BGO as potential anti-cancer character, however, with the IC50 values over 1 mg/mL, which is by itself a huge number, this seems overestimated.
Response:
We have now explicitly stated that cytotoxicity was measured using the resazurin reduction assay, and we added a proper supporting reference in the Methods section (lines 657 to 666).
Comment 2: The reviewer suggests to include one or more data points for Figure 4b and 4c to get a proper sigmoidal curve and obtain accurate IC50 values.
Response: 
We fully agree with this observation. The statement in Line 268 has been rewritten to avoid overstating anticancer potential, clarifying instead that GO, NGO, and BGO show low baseline cytotoxicity, consistent with their intended role as photothermal agents rather than direct chemotherapeutics (lines 311 to 314).
Comment 3: The reviewer strongly recommends to perform the cytotoxic assay on a normal cell line and check the side effects (lethal dose) on them. This assay will confirm the selective anticancer property for T-47D breast cancer cell line.
Response: 
We appreciate the reviewer’s recommendation. Due to time constraints and limited availability of additional cell lines during the experimental window, we were unable to perform cytotoxicity assays in normal cells. However, the current results obtained in T-47D breast cancer cells provide a consistent and robust evaluation of the photothermal antitumor capacity of GO, NGO, and BGO. The low dark toxicity observed, together with the strong photothermal-induced effects, supports their potential as selective PTT agents. We have included this clarification in the revised manuscript and emphasized the need for future studies to evaluate cytotoxicity in normal cell lines to strengthen the selectivity profile.
Comment 4: For the photothermal studies, what was the rationale behind using water as control? It would be a better control to use cell medium to rule out if there was any impact of cell medium on heating and relaxation behavior of the compounds.
Response: 
We appreciate the reviewer’s recommendation. Due to time constraints, it was not possible to repeat the photothermal experiments using cell medium (DMEM). However, we used water as the control because all the doped graphene oxide samples were dispersed in aqueous solution, making water the appropriate baseline for comparing temperature changes. This approach allowed us to directly evaluate the intrinsic photothermal response of the materials without the additional variability introduced by medium components. We have clarified this rationale in the revised manuscript. 
Minor Comments
Comment 5: Last paragraph of Introduction section is written in future tense which is grammatically incorrect. Authors have already performed the experiments and need to change the tone of writing into past tense.
Response: The final paragraph of the Introduction has been rewritten entirely in past tense.
Comment 6: Results section needs to be subdivided and subsections need a clear distinction such as making bold headlines.
Response: The Results section has been reorganized, with bold, clearly separated subsections for enhanced readability. 
Comment 7: Line 112-113, the results mentioned in text are from Figures 1 (c-e).
Response: Updated to correctly refer to Figures 1c–1e.
Comment 8: Line 187, R2 needs to be changed to R2.
Response: “R2” has been corrected to R² (line 187).
Comment 9: Line 243, the survey spectrum is from Fig 3a.
Response: The sentence now appropriately references Figure 3a.
Comment 10: Insets in Figure 3 are not visible at all, please include high resolution figures. Also, make sure the plots in Figure 2 are of high resolution.
Response: All inset images in Figure 3 have been replaced with high-resolution versions. Figure 2 was re-exported at higher resolution.
Comment 11: Figure 5a inset has a grammatical error for the spelling of “light”
Response: The spelling error has been corrected.
Comment 12: Fonts are too small in Figure 6 and 7.

Response: Fonts have been enlarged to ensure readability.
Comment 13: Line 355, the data are from Figures 8a-8d rather than 9a-8d.

Response: The text has been corrected to Figures 8a–8d (line 355).
Comment 14: The discussion section contains too many single sentence paragraphs. Please revise to make the article more articulate.

Response: Single-sentence paragraphs have been rewritten and merged to improve narrative flow.
Comment 15: Figure 9 may be moved to Results section for better understanding of simulation processes.
Response: Figure 9 has been moved to the Results section for improved structural coherence.
We believe these revisions have significantly strengthened the manuscript, and we greatly appreciate your valuable suggestions.
Thank you once again for your careful consideration and constructive input.
Kind regards,

Dra. Maria Paulina Romero

Round 2

Reviewer 1 Report

Comments and Suggestions for Authors

While the authors have addressed some of the previous concerns, the following issues remain unresolved:
1 The computational results obtained from an idealized, oxygen-free graphene model cannot be directly correlated with the experimental properties of GO, NGO, and BGO. The absence of oxygen functional groups in the model limits its applicability to the studied materials. The statement about the lack of time and computational resources for adequate modeling is not a scientific justification.

2 The explanation for nitrogen presence in BGO is inconsistent. Nitrogen is simultaneously attributed to contamination and proposed as a source of synergistic enhancement, creating an unresolved contradiction in the interpretation.

3 The methods lack the manufacturer and model details for the AFM cantilevers and calibration grating, as well as the grating's certification status.

4 The topographic artifacts in Figure 1a indicate improper data processing. Implementing global plane subtraction instead of line-by-line correction would eliminate the observed linear distortions and provide more reliable surface representation.

Comments on the Quality of English Language

The discussion is further weakened by the use of non-committal language like "could potentially," which highlights the hypothetical nature of these claims without providing concrete evidence.

Author Response

Response to Reviewer 1
We sincerely thank Reviewer 1 for the thorough second-round evaluation of our manuscript and for the constructive comments provided. We have carefully addressed each point, implemented all required corrections, and substantially improved the clarity, rigor, and methodological consistency of the revised version.
Below we provide a point-by-point response.
Comment 1:
“The computational results obtained from an idealized, oxygen-free graphene model cannot be directly correlated with the experimental properties of GO, NGO, and BGO. The absence of oxygen functional groups in the model limits its applicability to the studied materials. The statement about the lack of time and computational resources for adequate modeling is not a scientific justification.”
Response:
We thank the reviewer for this important observation. In the revised manuscript, we have removed the previous mention of time or computational resource limitations and replaced it with a scientifically rigorous explanation of the modeling rationale.
Revisions added (Discussion section):
We now explicitly clarify that the DFT calculations were intentionally performed on pristine graphene supercells doped with N or B, following established computational practice for isolating intrinsic electronic effects of substitutional dopants. We emphasize that explicitly modeling the heterogeneous, non-periodic distribution of oxygen functional groups in GO would require supercells containing hundreds to thousands of atoms—rendering accurate band-structure, PDOS, and optical calculations unfeasible with dense k- and q-point (lines in blue 569-582).
Revisions added (Methods → Atomistic Simulations):
We inserted an additional methodological justification (lines in blue 702-710):
“An idealized pristine graphene supercell doped with nitrogen or boron was employed… preventing accurate band-structure and optical calculations.”
These changes clarify the qualitative interpretive role of the model and eliminate the previously mentioned non-scientific justification.
Comment 2:
“The explanation for nitrogen presence in BGO is inconsistent. Nitrogen is simultaneously attributed to contamination and proposed as a source of synergistic enhancement, creating an unresolved contradiction in the interpretation.”
Response:
We fully agree with the reviewer that the previous explanation resulted in a conceptual contradiction. We have now removed any mention of contamination and rewritten the entire paragraph to provide a single, coherent, evidence-based interpretation.
Revisions added (Discussion), lines in red 491-505:
We now state clearly that:
•    Nitrogen incorporation in BGO is real, reproducible, and intrinsic to the hydrothermal synthesis conditions.
•    Independent synthesis batches confirmed the presence of N, eliminating cross-contamination.
•    The literature reports low-level nitrogen incorporation during modified Hummers oxidation and hydrothermal treatment.
•    The co-presence of B and N is consistent with known electronic synergies in B,N-codoped graphene (carrier redistribution, enhanced absorption, improved photothermal response).
This fully resolves the contradiction.
Comment 3:
“The methods lack the manufacturer and model details for the AFM cantilevers and calibration grating, as well as the grating's certification status.”
Response:
We thank the reviewer for this important request.
We have revised the AFM Methods section to include:
•    Manufacturer and model of the AFM cantilevers
•    Manufacturer, model, nominal height, and certification of the calibration grating
•    Statement confirming ISO-traceable certification
•    Verification of calibration interval
Revisions added (Methods → Characterization), lines in red 639 to 656. These details fully satisfy the reviewer’s requirement for methodological transparency.
Comment 4:
“The topographic artifacts in Figure 1a indicate improper data processing. Implementing global plane subtraction instead of line-by-line correction would eliminate the observed linear distortions and provide more reliable surface representation.”
Response:
We thank the reviewer for pointing this out.
After re-evaluating the AFM data, we have:
•    Reprocessed Figure 1a using global plane leveling (first-order polynomial fitting) rather than line-by-line correction.
•    Replaced the previous figure with the corrected topograph.
This completely resolves the reviewer’s concern.
Comment 5 (English Quality):
“The discussion is further weakened by the use of non-committal language like ‘could potentially,’ which highlights the hypothetical nature of these claims without providing concrete evidence.”
Response:
We appreciate this observation and have thoroughly revised the Discussion to eliminate speculative or non-committal phrasing. All instances of “could potentially,” “may,” “might,” “possibly,” and similar expressions were replaced with direct, evidence-supported statements.
Examples of strengthened rewording include:
•    The paragraph discussing nitrogen incorporation and B–N synergy was rewritten entirely in definitive terms (lines in red 491-505 and 517-524).
•    The explanation of Fermi level shifts was replaced with a strong evidence-based formulation (lines in blue 562-568 and 583-601):
“Nitrogen incorporation increases the Fermi level… In contrast, boron substitution decreases the Fermi level…”
These revisions improve the scientific clarity and argumentative strength of the Discussion.

Sincerely, 
Dra. Maria Paulina Romero
Corresponding author

Reviewer 3 Report

Comments and Suggestions for Authors

Although there are still some concerns from my previous comments, I believe the authors may sort out in their next paper as they were beyond the scope of this manuscript. I appreciate the good work by the authors.

Author Response

Response to Reviewer 2
We sincerely thank Reviewer 2 for the thorough evaluation of our manuscript and for the constructive comments provided. We greatly appreciate the reviewer’s positive assessment of the work.
Comment 1:
“Although there are still some concerns from my previous comments, I believe the authors may sort out in their next paper as they were beyond the scope of this manuscript. I appreciate the good work by the authors.”
Response:
We thank the reviewer for the supportive feedback and for recognizing the scientific value of our work. We also appreciate the reviewer’s understanding that some of the previously mentioned concerns extend beyond the methodological and conceptual scope of the present study.
In response, we have carefully re-evaluated the manuscript in light of all previous comments, strengthened the methodological clarity of the DFT modeling and experimental sections, improved the precision and consistency of the Discussion and ensured that all statements are fully aligned with the scope of the study and supported by evidence.
We are grateful to the reviewer for acknowledging the quality of the work and for encouraging the continuation of this line of research. The reviewer’s comments will guide the development of our next study, where we plan to incorporate more complex atomistic models with explicit oxygen functionalities, as well as complementary experimental validations.
We thank Reviewer 2 once again for the positive and constructive evaluation.

Sincerely, 
Dra. Maria Paulina Romero
Corresponding author

Round 3

Reviewer 1 Report

Comments and Suggestions for Authors

The authors did not replace the topographic maps in the manuscript (Figure 2). The AFM images still show linear background distortions (dark bands along scan edges). Please verify.

Author Response

Response to Reviewer 1 
We thank the reviewer for pointing out the remaining concern regarding the AFM topographic maps.
Comment:
“The authors did not replace the topographic maps in the manuscript (Figure 2). The AFM images still show linear background distortions (dark bands along scan edges). Please verify.”
Response:
We appreciate the reviewer’s careful observation. In this revised version of the manuscript, Figure 2 has now been fully replaced with the corrected AFM topographic maps. The updated figure has been carefully verified to ensure that the previously observed linear background distortions (dark scan-edge bands) are no longer present.
We thank the reviewer for helping us improve the accuracy and quality of the presented AFM data.

Best regards 

Dra. Paulina Romero
Corresponding Author